# WHY ADAM BEATS SGD FOR ATTENTION MODELS

## ABSTRACT

While stochastic gradient descent (SGD) is still the *de facto* algorithm in deep learning, adaptive methods like Adam have been observed to outperform SGD across important tasks, such as attention models. The settings under which SGD performs poorly in comparison to Adam are not well understood yet. In this paper, we provide empirical and theoretical evidence that a heavy-tailed distribution of the noise in stochastic gradients is a root cause of SGD's poor performance. Based on this observation, we study clipped variants of SGD that circumvent this issue; we then analyze their convergence under heavy-tailed noise. Furthermore, we develop a new *adaptive* coordinate-wise clipping algorithm (ACClip) tailored to such settings. Subsequently, we show how adaptive methods like Adam can be viewed through the lens of clipping, which helps us explain Adam's strong performance under heavy-tail noise settings. Finally, we show that the proposed ACClip outperforms Adam for both BERT pretraining and finetuning tasks.

## 1 INTRODUCTION

Stochastic gradient descent (SGD) is the canonical algorithm for training neural networks (Robbins & Monro, 1951). SGD iteratively updates model parameters in the negative gradient direction and thus seamlessly scales to large-scale settings. Though a well-tuned SGD outperforms adaptive methods (Wilson et al., 2017) in many traditional tasks including ImageNet classification (see Figure 1b), certain tasks necessitate the use of *adaptive* variants of SGD (e.g., Adagrad (Duchi et al., 2011), Adam (Kingma & Ba, 2014), AMSGrad (Reddi et al., 2019)), which employ adaptive per-parameter learning rates. For instance, consider the task of training an attention model (Vaswani et al., 2017) using BERT (Devlin et al., 2018). Figure 1a shows loss curves of BERT pretraining obtained from SGD with momentum as well as Adam. It can be seen that in spite of extensive hyperparameter tuning, SGD converges much slower than Adam during BERT training.

The first significant hint to the performance of Adam on BERT comes from the distribution of the stochastic gradients. For Imagenet, the norms of the mini-batch gradients are typically quite small and well concentrated around their mean. On the other hand, the mini-batch gradient norms for BERT take a wide range of values and are sometimes much larger than their mean value. More formally, while the distribution of the stochastic gradients in Imagenet is well approximated by a Gaussian, the distribution for BERT seems to be *heavy-tailed*.

In this work, we perform a rigorous theoretical and empirical study of the convergence of optimization methods under such heavy-tailed noise. In this setting, some of the stochastic gradients are much larger than the mean and can excessively influence the updates of SGD. This makes SGD unstable and leads to its poor performance. A natural strategy to stabilize the updates is to *clip* the magnitude of the stochastic gradients. We prove that indeed this is sufficient to ensure convergence even under heavy-tailed noise. Further, we show that adaptive methods in fact implicitly have such a clipping behavior, thereby providing a explanation for their superiority to SGD on BERT.

More specifically, we make the following main contributions:

- We empirically show that in tasks on which Adam outperforms SGD (BERT pretraining), the noise in stochastic gradients is heavy-tailed. On the other hand, on tasks where traditionally SGD outperforms Adam (ImageNet training), we show that the noise is well concentrated.

- We study the convergence of gradient methods under heavy-tailed noise condition where SGD's performance degrades and previous convergence proofs fail. We then establish the convergence

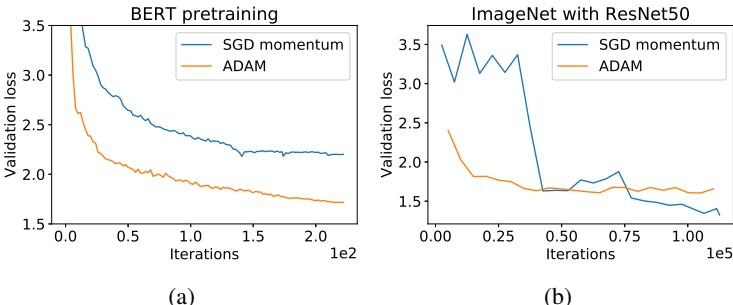

Figure 1: (a) Validation loss for BERT$_{base}$ pretraining. Although hyperparameters for SGD momentum are finetuned, a large performance gap is still observed between SGD and Adam. (b) Validation loss for ResNet50 trained on ImageNet. SGD momentum outperforms Adam

of *clipped* gradient methods under the same condition and prove that they obtain theoretically *optimal* rates.

- We show that Adam implicitly performs coordinate-wise gradient clipping and can hence, unlike SGD, tackle heavy-tailed noise. We prove that using such coordinate-wise clipping thresholds can be significantly faster than using a single global one. This can explain the superior performance of Adam on BERT pretraining.

- Inspired by our theoretical analysis, we propose a novel *adaptive*-threshold coordinate-wise clipping algorithm and experimentally show that it outperforms Adam on BERT training tasks.

## 1.1 RELATED WORK

**Adaptive step sizes.** Adaptive step size during optimization has long been studied (Armijo, 1966; Polyak, 1987). More recently, Duchi et al. (2011) developed the Adagrad algorithm that benefits from the sparsity in stochastic gradients. Inspired by Adagrad, several adaptive methods have been proposed in the deep learning community (Tieleman & Hinton, 2012; Kingma & Ba, 2014). Recently, there has been a surge in interest to study the theoretical properties of these adaptive gradient methods due to (Reddi et al., 2019), which pointed out the non-convergence of Adam and proposed an alternative algorithm, AMSGrad. Since then, many works studied different interesting aspects of adaptive methods, see (Ward et al., 2018; Li & Orabona, 2018; Zhou et al., 2018a; Staib et al., 2019; Chen et al., 2018; Zou & Shen, 2018; Zhou et al., 2018b; Agarwal et al., 2018; Zou et al., 2019). Another direction of related work is normalized gradient descent, which has been studied for quasi-convex and non-convex settings (Levy, 2016; Hazan et al., 2015; Zhang et al., 2019). In contrast to our work, these prior works assume standard noise distributions that might not be applicable to key modern applications such as attention models, which exhibit heavy-tailed noise. Furthermore, mostly, the convergence rates of adaptive methods are worse than SGD.

**Noise in neural network.** There has been little study of the actual stochastic gradient noise distributions in neural network training. To our knowledge, Simsekli et al. (2019) is the first work to focus on this topic and observe heavy tailed noise in network training. Our work differs in two important ways: *first*, we treat the noise as a high dimensional vector, while (Simsekli et al., 2019) treat deviations in each coordinate as scaler noises to estimate tail index. *Second*, we focus on convergence of optimization algorithm, the original paper focus on Langevin dynamics and consistence of the index estimator. More experiment comparison is in Appendix H.

## 2 HEAVY-TAILED NOISE IN STOCHASTIC GRADIENTS

To gain intuition about the difference between SGD and Adam, we start our discussion with the study of noise distributions of stochastic gradient that arise during neural network training. In particular, we focus on noise distributions while training two popular deep learning models — BERT and ResNet. Note that BERT and ResNet are typically trained with Adam and SGD (with momentum) respectively and can thus, provide insights about difference between these optimizers.

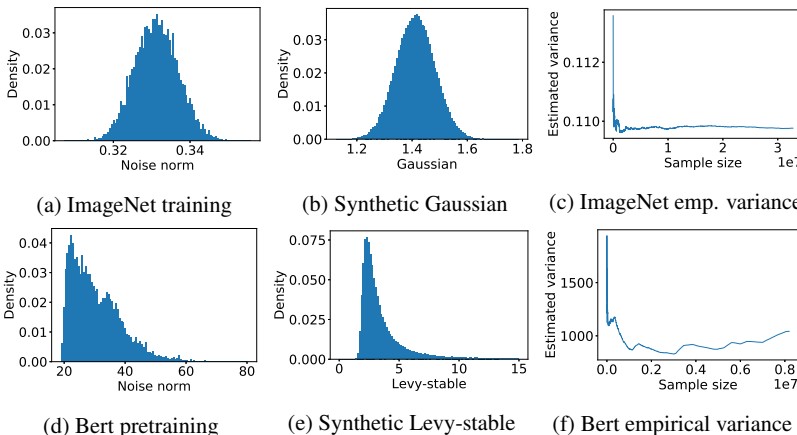

Figure 2: (a) Histogram of sampled gradient noise for ResNet50 using Imagenet dataset. (b) Histogram of samples from a sum of squared Gaussians. (c) Estimated variance of the stochastic gradient for Resnet50. (d) Histogram of sampled gradient nosie for BERT using Wikipedia+Books dataset. (e) Histogram of samples from a sum of squared $\alpha$-stable random variables. (f) Estimated variance of the stochastic gradient for BERT model.

We first investigate the distribution of the gradient noise norm $\|g - \nabla f(x)\|$ in the aforementioned neural network models, where $g$ is the stochastic gradient computed from a minibatch sample. Figure 7 (a) and (d) show these distributions for ResNet50 on ImageNet and BERT on the Wikipedia and books dataset at model initialization respectively. For comparison, we plot distributions of a normalized sum of squared Gaussians, a well-concentrated distribution, and a Levy-$\alpha$-stable distribution, a heavy-tailed distribution, in Figure 7 (b) and (e) respectively. We observe that the noise distribution for BERT appears heavy-tailed, while that of ResNet50 is well-concentrated.

To support this observation, in Figure 7 (c) and (f) we further show the empirical variance of stochastic gradients with respect to the sample size used in the estimation. The results highlight that while the corresponding estimator converges for Imagenet, the empirical variance does not converge in BERT training even as the sample size approaches $10^7$.

Figure 7 clearly shows that while the noise pattern is well-concentrated in some cases, it is also heavy tailed in other settings. We hypothesize that this is one major aspect that determines the performance of SGD and adaptive methods. More specifically, we argue and provide evidence that adaptive methods can be faster than SGD in scenarios where heavy-tailed noise distributions arise. More empirical details can be found in Section 5.

## 2.1 Optimization problem setup

We now formalize the optimization setup with heavy-tailed noise. Neural network training can be seen as minimizing a differentiable stochastic function $f(x) = \mathbb{E}_\xi[f(x, \xi)]$, where $f : \mathbb{R}^d \to \mathbb{R}$ can be potentially nonconvex. At each iteration, we assume access to an *unbiased* stochastic gradient $g(x) = \nabla f(x, \xi)$ corresponding to the parameters $x$, and also assume finite $\alpha$ moment.

**Assumption 1 (Existence of $\alpha-$moment).** *There exists positive real numbers $\alpha \in (1, 2]$ and $G > 0$ such that for all $x$, $\mathbb{E}[\|g(x)\|^\alpha] \leq G^\alpha$.*

Note that it is possible that the variance of $g(x)$ is unbounded while simultaneously satisfying the above assumption for $\alpha < 2$, e.g. the Pareto distribution. Hence Assumption 1 is much weaker than the standard bounded second moment assumption. The possibility that the variance could be unbounded has a profound impact on the optimization process. In particular, previous convergence proofs for SGD fail and the algorithm may diverge (see Appendix A). As we will illustrate over the next few sections, using a biased *clipped* stochastic gradient estimator allows us to circumvent the problem of unbounded variance, while ensuring convergence even under heavy-tailed noise.

## 3 CLIPPED GRADIENT DESCENT AND ACCLIP ALGORITHM

Our key idea is that by *clipping* the stochastic gradient, we can effectively control its variance. While this does introduce some additional bias, the clipping threshold can be (possibly adaptively) tuned to trade-off variance with the introduced bias.

Empirically, this is a prevalent practice e.g. Merity et al. (2018) use clipped SGD to train AWD-LSTM whereas Devlin et al. (2018) use clipped Adam to pretrain BERT. The general form of clipped gradient descent is outlined in Algorithm 1.

---

**Algorithm 1** A general framework for clipped gradient descent

---

1: $x, m_k \leftarrow x_0, 0$
2: **for** $k = 1, \cdot, T$ **do**
3: $\quad m_k \leftarrow \beta_1 m_{k-1} + (1 - \beta_1)g_k$
4: $\quad \hat{g}_k \leftarrow \text{clip}(\tau_k, m_k)$
5: $\quad x_k \leftarrow x_{k-1} - \eta_k \hat{g}_k$
$\quad$ **return** $x_K$, where random variable $K$ is supported on $\{1, \cdots, T\}$.

---

Perhaps the two most natural candidates for clipping mechanisms are Global CLIPping where a single global threshold is used and Coordinate-wise CLIPping using $d$ coordinate-wise thresholds

$$\text{GClip}(\tau_k, m_k) = \min\left\{\frac{\tau_k}{\|m_k\|}, 1\right\}m_k, \text{ for } \tau_k \in \mathbb{R}_{\geq 0} \text{ or,} \tag{GClip}$$

$$\text{CClip}(\tau_k, m_k) = \min\left\{\frac{\tau_k}{|m_k|}, 1\right\}m_k, \text{ for } \tau_k \in \mathbb{R}_{\geq 0}^d. \tag{CClip}$$

where all operations for CClip are performed element-wise. Smaller threshold values of $\tau_k$ imply smaller variance, but also higher bias. If the noise distribution is very heavy-tailed (or has a large variance), one could compensate by picking a smaller threshold. See Theorems 1 and 2 for a theoretical treatment of this for GClip. While GClip preserves the update direction and only scales its magnitude, CClip scales each coordinate individually and may not preserve the direction. However, if the noise distribution varies significantly across coordinates, CClip can take advantage of this and only clip those which are more heavy-tailed and consequently converge faster (see Theorem 3).

However, to perform such a coordinate-wise variance-bias trade off optimally would require tuning all $d$ thresholds, which could be extremely large in deep learning. Even if tuning were feasible, the noise distribution may be non-stationary and significantly vary as training progresses (see Fig. 5). To address this challenge, we propose Adaptive Coordinate-wise CLIPping, which *adaptively* sets the thresholds for each coordinate

$$\text{ACClip}(\tau_k, m_k) = \min\left\{\frac{\tau_k}{|m_k|+\epsilon}, 1\right\}m_k, \quad \tau_k^\alpha = \beta_2\tau_{k-1}^\alpha + (1 - \beta_2)|g_k|^\alpha, \tag{ACClip}$$

where operations are element-wise, $\epsilon$ is a small number for numerical stability, and we set $\alpha = 1$ by default. The use of a coordinate-wise $\alpha$-moment (as opposed to the traditional second moment) is motivated by our theory and estimates $B = \mathbb{E}[|g_k^\alpha|]^{1/\alpha}$ in Assumption 2. Since $\mathbb{E}[|g_k^\alpha|]^{1/\alpha}$ is increasing in $\alpha \geq 1$, using $\alpha = 1$ is a conservative bound on $B$ when the true $\alpha$ is unknown.

### 3.1 ADAM AS AN ADAPTIVE CLIPPING METHOD

We study Adam without momentum, i.e. RMSProp, and show that it resembles coordinate-wise clipping. When $\beta_1 = 0$, we can rewrite the update for Adam and ACClip as SGD with effective step-sizes $h_{\text{Adam}}$ and $h_{\text{clip}}$ respectively

$$x_{k+1} = x_k - \frac{\alpha}{\epsilon + \sqrt{\beta_2 v_k + (1-\beta_2)|g_k|^2}}g_k =: x_k - h_{\text{Adam}}g_k, \text{ and}$$

$$x_{k+1} = x_k - \eta_k \min\left\{\frac{\tau_k}{|g_k|}, 1\right\}g_k =: x_k - h_{\text{clip}}g_k.$$

Given any set of parameters for RMSProp, if we set the parameters for ACClip as

$$\eta_k = \frac{2\alpha}{\epsilon + \sqrt{\beta_2 v_k}} \quad \text{and} \quad \tau_k = \frac{\epsilon + \sqrt{\beta_2 v_k}}{\sqrt{1-\beta_2}},$$

then $\frac{1}{2}h_{\text{clip}} \leq h_{\text{Adam}} \leq 2h_{\text{clip}}$. Thus, the two differ by at most af factor of 2 and Adam can be seen as ACClip where $\tau_k$ is set using $\sqrt{v_k}$, which estimates $\mathbb{E}[|g_k|^2]^{1/2}$, and a correspondingly decreasing

Table 1: Error bounds after $k$ iterations: Define $\alpha$-moment $\mathbb{E}[\|g(x)\|^{\alpha}] \leq G^{\alpha}$ (Assump 1) and coordinate-wise moments $\mathbb{E}[|g(x)|^{\alpha}] \leq B^{\alpha}$ (Assump 2), which satisfy $G^2 \leq d\|B\|_{\alpha}^2$. In the standard setting ($\alpha = 2$), GClip recovers the optimal rates of SGD. For heavy-tailed noise ($\alpha \in (1, 2)$), GClip converges both for convex (Thm 1) and non-convex functions (Thm 2), whereas the proof for SGD fails, denoted as N/A. Under a more fine-grained noise model, CClip has better convergence rates (Thm 3). We also show matching lower-bounds for all $\alpha \in (1, 2]$ proving the optimality of clipping methods (Thm 4).

| | Strongly Convex Function | | Non-Convex Function | |
|---|---|---|---|---|
| | Heavy-tailed noise ($\alpha \in (1, 2)$) | Standard noise ($\alpha \geq 2$) | Heavy-tailed noise ($\alpha \in (1, 2)$) | Standard noise ($\alpha \geq 2$) |
| SGD | N/A | $\mathcal{O}(G^2 k^{-1})$ | N/A | $\mathcal{O}(Gk^{-\frac{1}{2}})$ |
| GClip | $\mathcal{O}(G^2 k^{\frac{-2(\alpha-1)}{\alpha}})$ | $\mathcal{O}(G^2 k^{-1})$ | $\mathcal{O}(G^{\frac{2\alpha}{3\alpha-2}} k^{\frac{-2(\alpha-1)}{3\alpha-2}})$ | $\mathcal{O}(Gk^{-\frac{1}{2}})$ |
| CClip | $\mathcal{O}(\|B\|_2^2 k^{\frac{-2(\alpha-1)}{\alpha}})$ | $\mathcal{O}(\|B\|_2^2 k^{-1})$ | $\mathcal{O}(\|B\|_2^{\frac{2\alpha}{3\alpha-2}} k^{\frac{-2(\alpha-1)}{3\alpha-2}})$ | $\mathcal{O}(\|B\|_2 k^{-\frac{1}{2}})$ |
| Lower-bound | $\Omega(k^{\frac{-2(\alpha-1)}{\alpha}})$ | $\Omega(k^{-1})$ | ? | ? |

step-size. This clipping effect can explain the faster convergence of Adam for training attention models in the presence of heavy-tailed noise (Fig. 1a). In contrast to Adam, ACClip (with $\alpha = 1$) uses a simpler update scheme where $\tau_k$ is set using an estimate of $\mathbb{E}[|g_k|]$ and a constant step-size. This allows ACClip to take larger steps than Adam and hence converge slightly faster (Section 5).

## 4  THEORETICAL ANALYSIS

In this section, we address the following three key points. First, we show how gradient clipping ensures convergence for both convex and non-convex functions under heavy-tailed noise. Then, we study when and why coordinate-wise clipping (CClip) outperforms global clipping (GClip). Finally, we show that (upto numerical constants) clipping is an optimal strategy under the strongly convex setting. Together, these provide a comprehensive analysis of the effect of clipping and a strong theoretical motivation for using it whenever one suspects the presence of noise with unbounded variance. Our results are summarized in Table 1, and all proofs are relegated to the Appendices.

### 4.1  CONVERGENCE OF GLOBALLY CLIPPED SGD

We state the rates for strongly-convex functions and for smooth non-convex functions. We focus on GClip with momentum $\beta = 0$, under the assumption 1. In the strongly convex case, we can prove the following rates of convergence to the optimum.

**Theorem 1 (Strongly-convex convergence).** *Suppose that $f$ is a $\mu$-strongly convex function and the noise satisfies assumption 1. Then let $\{x_k\}$ be the iterates of Algorithm 1 using GClip, clipping parameter $\tau_k = Gk^{\alpha-1}$ and steps-size $\eta_k = \frac{4}{\mu(k+1)}$. Define the output to be a $j$-weighted combination of the iterates: $\bar{x}_k = \sum_{j=1}^{k} j x_{j-1} / (\sum_{j=1}^{k} j)$. Then the output $\bar{x}_k$ satisfies:*

$$\mathbb{E}[f(\bar{x}_k)] - f(x^{\star}) \leq \frac{16G^2}{\mu(k+1)^{2(\alpha-1)/\alpha}} .$$

Observe that when $\alpha = 2$ and with bounded second moment, we recover exactly the optimal SGD rate of $\mathcal{O}(G^2/\mu k)$ of Lacoste-Julien et al. (2012). For any other $\alpha \in (1, 2)$, the performance of GClip gracefully degrades. As we will later show, this rate is in fact optimal for every $\alpha \in (1, 2]$. In contrast, SGD can be arbitrarily bad when $\alpha < 2$ (see Appendix A). We can also prove rates of convergence to a stationary point for smooth *non-convex* functions.

**Theorem 2 (Non-convex convergence).** *Suppose that $f$ is a possible non-convex $L$-smooth function and the noise satisfies assumption 1. Then let $\{x_k\}$ be the iterates of Algorithm 1 using GClip with momentum $\beta = 0$, constant step-size $\eta_k = \left[\left(\frac{f(x_0)-f^{\star}}{G^2(K+1)}\right)^{\frac{\alpha}{3\alpha-2}} / L^{\frac{2\alpha-2}{3\alpha-2}}\right]$ and constant clipping parameter $\tau_k = G/(\eta L)^{\frac{1}{\alpha}}$. Then, the sequence $\{x_k\}$ satisfies*

$$\frac{1}{K}\sum_{k=1}^{K}\mathbb{E}[\|\nabla f(x_{k-1})\|^2] \leq 4G^{\frac{2\alpha}{3\alpha-2}}\left(\frac{f(x_0)-f(x^{\star})}{K}\right)^{\frac{2\alpha-2}{3\alpha-2}}.$$

The rates of convergence for the non-convex case in Theorem 2 exactly match those of the usual SGD rates of $\mathcal{O}(1/\sqrt{k})$ when $\alpha = 2$ and gracefully degrades for $\alpha \in (1,2]$. Thus, GClip converges for both convex and non-convex functions under Assumption 1. However, Assumption 1 is quite crude and summarizes noise into a single number $G$. The noise present in the different gradient coordinates can be quite varied, and as we next show, CClip can take advantage of this and clip each coordinate independently. This allows CClip (when tuned properly) to have a significantly better convergence rate when compared to GClip.

## 4.2 BENEFITS OF COORDINATE-WISE CLIPPING

We had previously assumed a global bound on the $\alpha$-moment of the *norm* of the stochastic gradient is bounded by $G$. However, $G$ might be hiding some dimension dependence $d$. We next study a more fine-grained model of the noise in order to tease out this dependence.

**Assumption 2 (Coordinate-wise $\alpha$ moment).** *Denote $\{g_i(x)\}$ to be the coordinate-wise gradients for $i \in [d]$. We assume there exist constants $\{B_i\} \geq 0$ and $\alpha \in (1,2]$ such that $\mathbb{E}[|g_i(x)|^{\alpha}] \leq B_i^{\alpha}$.*

For the sake of convenience, we denote $B = [B_1; B_2; \cdots ; B_d] \in \mathbb{R}^d$, $\|B\|_a = (\sum B_i^a)^{1/a}$. Under this more refined assumption, we can show the following corollary:

**Corollary 1 (GClip under coordinate-wise noise).** *Suppose we run GClip under the Assumption of 2 to obtain the sequence $\{x_k\}$. Then, if $f$ is $\mu$-strongly convex, with appropriate step-sizes and averaging, the output $\bar{x}_k$ satisfies*

$$\mathbb{E}[f(\bar{x}_k)] - f(x^{\star}) \leq \frac{16d\|B\|_{\alpha}^2}{\mu(k+1)^{2(\alpha-1)/\alpha}}.$$

Thus, the convergence of GClip can have a pretty strong dependence on $d$, which for large-scale problems might be problematic. We show next that using coordinate-wise clipping avoids this issue.

**Theorem 3 (CClip under coordinate-wise noise).** *Suppose we run CClip under the Assumption of 2 with $\tau_k = Bk^{\alpha-1}$ to obtain the sequence $\{x_k\}$. Then, if $f$ is $\mu$-strongly convex, with appropriate step-sizes and averaging, the output $\bar{x}_k$ satisfies*

$$\mathbb{E}[f(\bar{x}_k)] - f(x^{\star}) \leq \frac{16\|B\|_2^2}{\mu(k+1)^{2(\alpha-1)/\alpha}}.$$

Note that $\|B\|_2 \leq \|B\|_{\alpha}$. Hence CClip has a worst-case convergence independent of $d$ under the coordinate-wise noise model. Similar comparison between GClip and CClip can be done for non-convex conditions too (see Thms 5 and 3 in Appendix F). This improvement has two sources: coordinate-wise clipping can adapt to different $B_i$ in each coordinate; the global $L_2$ clipping does not align with the geometry of coordinate-wise noise. Although we only compare upper-bounds here, when the noise across coordinates is independent the upper bounds here may be tight (see Lemma 4). Further, our experiments where $d \gg n$ (over-parameterized models) also empirically confirms superior performance of coordinate-wise clipping.

Given that CClip may be better than GClip, we next address the natural question of whether there exists some other strategy (which may perhaps not involve clipping at all) which is even better.

## 4.3 AN INFORMATION THEORETIC LOWER BOUND

We show a strong lower-bound for a simple class of one-dimensional quadratic functions with stochastic gradients satisfying $\mathbb{E}[|g(x)|^{\alpha}] \leq 1$. This matches the upper bounds of Theorems 1 and 3 for strongly-convex functions, showing that the simple clipping mechanism of Algorithm 1 is (up to constants) information theoretically optimal, providing a strong justification for its use.

**Theorem 4.** *For any $\alpha \in (1,2]$ and any (possibly randomized) algorithm $\mathcal{A}$, there exists a problem $f$ which is 1-strongly convex and 1-smooth ($\mu = 1$ and $L = 1$), and stochastic gradients which satisfy Assumptions 1 and 2 with $G, \|B\| \leq 1$ such that the output $x_k$ of the algorithm $\mathcal{A}$ after processing $k$ stochastic gradients has an error*

$$\mathbb{E}[f(x_k)] - f(x^{\star}) \geq \Omega\left(\frac{1}{k^{2(\alpha-1)/\alpha}}\right).$$

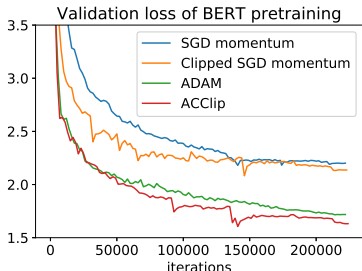

Figure 3: Validation loss for $\text{BERT}_{base}$ pre-training with the sequence length of 128. While there remains a large gap between non-adaptive methods and adaptive methods, clipped SGD momentum achieves faster convergence compared to standard SGD momentum. The proposed algorithm for adaptive coordinate-wise clipping (ACClip) achieves a lower loss than Adam.

Table 2: **BERT pretraining: Adam vs ACClip.** Compared to Adam, the proposed ACClip algorithm achieves better evaluation loss and Masked LM accuracy for all model sizes.

|  | BERT Base 6 layers | | BERT Base 12 layers | | BERT Large 24 layers | |
|---|---|---|---|---|---|---|
|  | Val. loss | Accuracy | Val. loss | Accuracy | Val. loss | Accuracy |
| Adam | 1.907 | 63.45 | 1.718 | 66.44 | 1.432 | 70.56 |
| ACClip | **1.877** | **63.85** | **1.615** | **67.16** | **1.413** | **70.97** |

## 5 EXPERIMENTS

We perform extensive evaluations of ACClip on BERT pre-training and fine-tuning tasks and demonstrate its advantage over Adam in Section 5.1. Since ACClip was explicitly designed to tackle heavy-tailed noise, this provides further evidence that Adam owes its efficacy to its gradient clipping effect. For completeness, an experiment on ImageNet is included in Appendix I. Then, in Section 5.2 we study the cause of the heavy-tailedness and show that both the architecture and the data play a role.

### 5.1 PERFORMANCE OF ACCLIP FOR BERT PRE-TRAINING AND FINE-TUNING

We evaluate the empirical performance of our proposed ACClip algorithm on BERT pre-training as well fine-tuning using the SQUAD v1.1 dataset. As a baseline, we use Adam optimizer and the same training setup as in the BERT paper (Devlin et al., 2018). For ACClip, we set $lr = 1 \times 10^{-4}, \beta_1 = 0.9, \beta_2 = 0.99, \epsilon = 1 \times 10^{-5}$ and $wd = 1 \times 10^{-5}$. We compare both setups on BERT models of three different sizes, $\text{BERT}_{base}$ with 6 and 12 layers as well as $\text{BERT}_{large}$ with 24 layers.

Figure 3 shows the validation loss for pretraining $\text{BERT}_{base}$ using SGD with momentum, GClip, Adam and ACClip. The learning rates and hyperparameters for each method have been extensively tuned to provide best performance on validation set. However, even after extensive tuning, there remains a large gap between (clipped) SGD momentum and adaptive methods. Furthermore, clipped SGD achieves faster convergence as well as lower final loss compared to standard SGD. Lastly, the proposed optimizer ACClip achieves a lower loss than the Adam. Table 2 further shows that ACClip achieves lower loss and higher masked-LM accuracy for all model sizes.

Next, we evaluate ACClip on the SQUAD v1.1 fine-tuning task. We again follow the procedure outlined in (Devlin et al., 2018) and present the results on the Dev set in Table 3. Both for F1 as well as for exact match, the proposed algorithm outperforms Adam on all model sizes.

The experimental results on BERT pretraining and fine-tuning indicate the effectiveness of the proposed algorithm. The increasing clipping threshold during the training is motivated by our theoretical analysis an indeed helps with empirical performance by reducing the bias in gradient estimators.

### 5.2 NOISE PATTERNS IN NEURAL NETWORK TRAINING

In the following experiment, we aim to understand the effect of model architecture and training data on the shape of gradient noise, and to understand how this shape evolves during training.

To address the first question, we measure the noise distribution in an Attention and a ResNet-like model on both Wikipedia and synthetic Gaussian data. We used $\text{BERT}_{base}$ model as the Attention model, and the ResNet is constructed by removing the self-attention modules in the transformer blocks. Gaussian synthetic data is generated by replacing the token embedding layer with normalized Gaussian input. The resulting noise histograms are plotted in Figure 4. From the figure,

Table 3: **SQUAD v1.1 dev set: Adam vs ACClip**. The mean and standard deviation of F1 and exact match score for 5 runs. The first row contains results reported from the original BERT paper, which are obtained by picking the best ones out of 10 repeated experiments.

| | BERT Base 6 layers | | BERT Base 12 layers | | BERT Large 24 layers | |
| --- | --- | --- | --- | --- | --- | --- |
| | EM | F1 | EM | F1 | EM | F1 |
| Adam (Devlin et al., 2018) | | | 80.8 | 88.5 | 84.1 | 90.9 |
| Adam | $76.85 \pm 0.34$ | $84.79 \pm 0.33$ | $81.42 \pm 0.16$ | $88.61 \pm 0.11$ | $83.94 \pm 0.19$ | $90.87 \pm 0.12$ |
| ACClip | $\mathbf{78.07 \pm 0.24}$ | $\mathbf{85.87 \pm 0.13}$ | $\mathbf{81.62 \pm 0.18}$ | $\mathbf{88.82 \pm 0.10}$ | $\mathbf{84.93 \pm 0.29}$ | $\mathbf{91.40 \pm 0.15}$ |

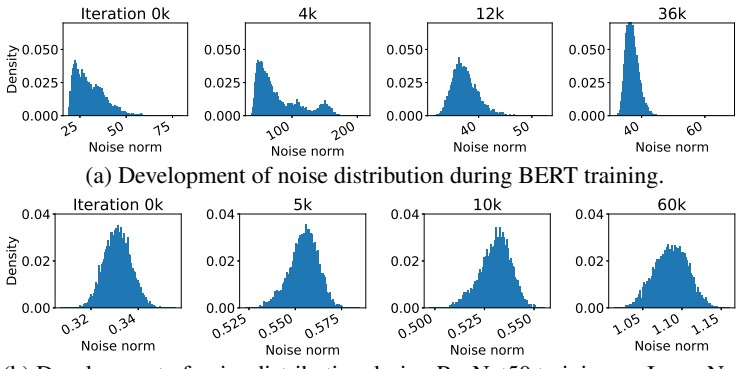

(a) Attention + Wikipedia  (b) Attention + Gaussian  (c) Resnet + Wikipedia  (d) Resnet + Gaussian.

Figure 4: Noise distribution in Attention and ResNet on two data sources: Wikipedia and synthetic Gaussian. The noise pattern is from the interaction of both architecture and data distribution.

heavy-tailed noises are observed in the Attention model independently of the input data. For the ResNet model, we see Gaussian input leads to Gaussian noise while Wikipedia data makes the noise to be heavy-tailed. Therefore, the noise pattern results from both the model architecture as well as the data distribution.

Figure 5 shows how the distribution of gradient noise is evolving during training. The results highlight that the noise distribution is non-stationary during BERT training as the noise is becoming increasingly more concentrated. In contrast, for the ResNet model on ImageNet, the shape of the noise distribution remains almost unchanged. This result supports the use of exponential moving average as an estimator for the underlying non-stationary distribution to accelerate optimization.

(a) Development of noise distribution during BERT training.

(b) Development of noise distribution during ResNet50 training on ImageNet.

Figure 5: The distribution of gradient noise is non-stationary during BERT training, while it remains almost unchanged for ResNet training on ImageNet.

## 6 CONCLUSION

Our work theoretically and empirically ties the advantage of adaptive methods over SGD to the heavy-tailed nature of gradient noise. A careful analysis of the noise and its impact yielded several insights: that adaptive methods implicitly perform gradient clipping, that clipping is an excellent strategy to deal with heavy-tailed noise, and that the adaptive coordinate-wise clipping yields state of the art performance for training attention models. Our results add to a growing body of work which demonstrate the importance of the structure of the noise in understanding neural network training. We believe additional such investigations into the source of the heavy tailed-ness, as well as a characterization of the noise in other scenarios where Adam is used (e.g. GAN training or reinforcement learning) can lead to further insights with significant impact on practice.

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

## A   NECESSITY OF ADAPTIVITY AND NON-CONVERGENCE OF SGD

Consider a simple toy function $f(x) = \frac{1}{2}\|x - a\|^2$ with an optimum value of 0 achieved at $x^\star = a$. Suppose that we receive stochastic gradients $g(x)$ with potentially unbounded variance i.e. $\mathbb{E}[\|g(x) - \nabla f(x)\|^2] = \infty$. Then, starting from $x_0 = 0$, let us run SGD (possibly with momentum) using with arbitrary but fixed step-sizes i.e. the step-sizes are predetermined and do not depend on the stochastic gradients. The output $x_k$ is then simply a linear combination of $\{g(x_j)\}_{j \leq k}$ with some (fixed) weights $\{\eta_j \in \mathbb{R}\}_{j \leq k}$

$$x_k = \sum_{j=1}^{k} \eta_j g(x_j).$$

The the expected function suboptimality at $x_k$ can be lower bounded as follows

$$\mathbb{E}[f(x_k)] - f(a) = \mathbb{E}[\|x_k - a\|^2] \geq \mathbb{E}[\|x_k - \mathbb{E}[x_k]\|^2] = \mathbb{E}\left[\left\|\sum_{j=1}^{k} \eta_j(g(x_j) - \mathbb{E}[g(x_j)])\right\|^2\right].$$

The last term in the above expression is the variance of sum of $k$ independent random variables and $\eta_j$ are fixed constants. We can then simplify as

$$\mathbb{E}\left[\left\|\sum_{j=1}^{k} \eta_j(g(x_j) - \mathbb{E}[g(x_j)])\right\|^2\right] = \sum_{j=1}^{k} \eta_j^2 \mathbb{E}\left[\|(g(x_j) - \mathbb{E}[g(x_j)])\|^2\right]$$

$$\geq (\textstyle\sum_{i=0}^{k} \eta_i^2) \max_{j \in [k]} \mathbb{E}[\|g(x_j) - \nabla f(x_j)\|^2].$$

This implies that if there is any $i, j \leq k$ such that $\eta_i^2 > 0$ and $\mathbb{E}[\|g(x_j) - \nabla f(x_j)\|^2] = \infty$, we have $\mathbb{E}[f(x_k)] = \infty$! Thus, the solution provided by SGD can be arbitrarily bad when the domain is unbounded.

It was important in our proof above that the step-sizes are fixed before-hand and are independent of the stochastic gradients. Notably, both gradient clipping and adaptive methods do not satisfy this property, and can indeed tackle unbounded variance. In some sense, the above example establishes adaptivity as being *necessary* when dealing with heavy-tailed noise.

## B   ADDITIONAL DEFINITIONS AND NOTATION

Here we describe some of the formal definitions which were previously skipped.

**Definition 3** ($\mu$-**strong-convexity**).  *$f$ is $\mu$-strongly convex, if there exist positive constants $\mu$ such that $\forall x, y$,*

$$f(y) \geq f(x) + \langle \nabla f(x), y - x \rangle + \frac{\mu}{2}\|y - x\|^2$$

The strong convexity assumption and the bounded gradient assumption implicitly implies that the domain is bounded. However, the domain diameter is not used explicitly in the proof. The projected clipped gradient descent follows exactly the same argument and the fact that orthogonal projections contract distances.

**Definition 4** ($L$-**smoothness**).  *$f$ is $L$-smooth, if there exist positive constants $L$ such that $\forall x, y$,*

$$f(y) \leq f(x) + \langle \nabla f(x), y - x \rangle + \frac{L}{2}\|y - x\|^2.$$

Note that the above definitions together imply that $\mu \leq L$.

## C   EFFECT OF GLOBAL CLIPPING ON VARIANCE AND BIAS

We focus on (GClip) under stochastic gradients which satisfy Assumption 1.

**Lemma 2.** *For any $g(x)$ suppose that assumption 1 holds with $\alpha \in (1, 2]$. Then the estimator $\hat{g}$ from (GClip) with global clipping parameter $\tau \geq 0$ satisfies:*

$$\mathbb{E}\left[\|\hat{g}(x)\|^2\right] \leq G^\alpha \tau^{2-\alpha} \text{ and } \|\mathbb{E}[\hat{g}(x)] - \nabla f(x)\|^2 \leq G^{2\alpha} \tau^{-2(\alpha-1)}.$$

*Proof.* First, we bound the variance.

$$\mathbb{E}[\|\hat{g}(x)\|^2] = \mathbb{E}[\|\hat{g}(x)\|^\alpha \|\hat{g}(x)\|^{2-\alpha}]$$

By the fact that $\hat{g}(x) \le \tau$, we get

$$\mathbb{E}[\|\hat{g}(x)\|^2] = \mathbb{E}[\|\hat{g}(x)\|^\alpha \tau^{2-\alpha}] \le G^\alpha \tau^{2-\alpha}.$$

Next, we bound the bias,

$$\begin{aligned}
\|\mathbb{E}[\hat{g}(x)] - \nabla f(x)\| &= \|\mathbb{E}[\hat{g}(x) - g(x)]\| \\
&\le \mathbb{E}[\|\hat{g}(x) - g(x)\|] = \mathbb{E}[\|\hat{g}(x) - g(x)\|\mathbb{1}_{\{|g(x)|\ge\tau\}}] \\
&\le \mathbb{E}[\|g(x)\|\mathbb{1}_{\{|g(x)|\ge\tau\}}] \le \mathbb{E}[\|g(x)\|^\alpha]\tau^{-(\alpha-1)}.
\end{aligned}$$

The last inequality follows by Markov inequality. $\qquad\square$

As we increase the clipping parameter $\tau$, note that the variance (the first term in Lemma 2) increases while the bias (which is the second term) decreases. This way, we can carefully trade-off the variance of our estimator against its bias, thereby ensuring convergence of the algorithm.

## D  STRONGLY-CONVEX RATES (PROOF OF THEOREM 1)

For simplicity, we denote $\hat{g}_k = \eta_k \hat{g}(x_k)$ and the bias $b_k = \mathbb{E}[\hat{g}_k] - \nabla f(x_k)$.

$$\begin{aligned}
\|x_k - x^*\|^2 &= \|x_{k-1} - \eta_k \hat{g}_{k-1} - x^*\|^2 \\
&= \|x_{k-1} - x^*\|^2 - 2\eta_k\langle x_{k-1} - x^*, \nabla f(x_{k-1})\rangle \\
&\quad - 2\eta_k\langle x_{k-1} - x^*, b_{k-1}\rangle + \eta_k^2\|\hat{g}_{k-1}\|^2 \\
&\le (1 - \mu\eta_k)\|x_{k-1} - x^*\|^2 - 2\eta_k(f(x_{k-1}) - f^*)) \\
&\quad + 2\eta_k(\frac{\mu}{4}\|x_{k-1} - x^*\|^2 + \frac{4}{\mu}\|b_k\|^2) + \eta_k^2\|\hat{g}_{k-1}\|^2.
\end{aligned}$$

Rearrange and we get

$$f(x_{k-1}) - f^* \le \frac{\eta_k^{-1} - \mu/2}{2}\|x_{k-1} - x^*\|^2 - \frac{\eta_k^{-1}}{2}\|x_k - x^*\|^2 + \frac{4}{\mu}\|b_k\|^2 + \frac{\eta_k}{2}\|\hat{g}_{k-1}\|^2.$$

After taking expectation and apply the inequality from Lemma 2, we get

$$\begin{aligned}
\mathbb{E}[f(x_{k-1})] - f^* &\le \mathbb{E}\left[\frac{\eta_k^{-1} - \mu/2}{2}\|x_{k-1} - x^*\|^2 - \frac{\eta_k^{-1}}{2}\|x_k - x^*\|^2\right] \\
&\quad + 4G^{2\alpha}\tau^{2-2\alpha}\mu^{-1} + \eta_k G^\alpha \tau^{2-\alpha}/2.
\end{aligned}$$

Then take $\eta_k = \frac{4}{\mu(k+1)}, \tau_k = Gk^{\frac{1}{\alpha}}$ and multiply both side by $k$, we get

$$\begin{aligned}
k\mathbb{E}[f(x_{k-1})] - f^* &\le \frac{\mu}{8}\mathbb{E}[k(k-1)\|x_{k-1} - x^*\|^2 - k(k+1)\|x_k - x^*\|^2] \\
&\quad + 8G^2 k^{\frac{2-\alpha}{\alpha}}\mu^{-1}.
\end{aligned}$$

Notice that $\sum_{k=1}^{K} k^{\frac{2-\alpha}{\alpha}} \le \int_0^{K+1} k^{\frac{2-\alpha}{\alpha}} dk \le (K+1)^{2/\alpha}$. Sum over $k$ and we get

$$\begin{aligned}
\sum_{k=1}^{K} k\mathbb{E}[f(x_{k-1})] - f^* &\le \frac{\mu}{8}\mathbb{E}[-T(T+1)\|x_T - x^*\|^2] \\
&\quad + 8G^2(K+1)^{\frac{2}{\alpha}}\mu^{-1}.
\end{aligned}$$

Devide both side by $\frac{K(K+1)}{2}$ and we get

$$\frac{2}{K(K+1)}\sum_{k=1}^{K} k\mathbb{E}[f(x_{k-1})] - f^* \le 8G^2 K^{-1}(K+1)^{\frac{2-\alpha}{\alpha}}\mu^{-1}.$$

Notice that for $K \ge 1$, $K^{-1} \le 2(K+1)^{-1}$. We have

$$\frac{2}{K(K+1)}\sum_{k=1}^{K} k\mathbb{E}[f(x_{k-1})] - f^* \le 16G^2(K+1)^{\frac{2-2\alpha}{\alpha}}\mu^{-1}.$$

The theorem then follows by Jensen's inequality. $\qquad\square$

## E    NON-CONVEX RATES (PROOF OF THEOREM 2)

For simplicity, we denote $\hat{g}_k = \hat{g}(x_{k-1})$ and the bias $b_k = \mathbb{E}[\hat{g}_k] - \nabla f(x_k)$. By Assumption 4, we have

$$f(x_k) \leq f(x_{k-1}) + \langle \nabla f(x_{k-1}), -\eta_k \hat{g}_k \rangle + \frac{\eta_k^2 L}{2} \|\hat{g}_k\|^2$$

$$\leq f(x_{k-1}) - \eta_k \|\nabla f(x_{k-1})\|^2 - \eta_k \langle \nabla f(x_{k-1}), b_k \rangle + \frac{\eta_k^2 L}{2} \|\hat{g}_k\|^2$$

$$\leq f(x_{k-1}) - \eta_k \|\nabla f(x_{k-1})\|^2 - \eta_k \langle \nabla f(x_{k-1}), b_k \rangle + \frac{\eta_k^2 L}{2} \|\hat{g}_k\|^2$$

$$\leq f(x_{k-1}) - \eta_k \|\nabla f(x_{k-1})\|^2 + \frac{\eta_k}{2} \|\nabla f(x_{k-1})\|^2 + \frac{\eta_k}{2} \|b_k\|^2 + \frac{\eta_k^2 L}{2} \|\hat{g}_k\|^2 .$$

Here the last step used the AM-GM inequality. Then, taking expectation in both sides and using Lemma 2 gives

$$\mathbb{E}[f(x_k)|x_{k-1}] \leq f(x_{k-1}) - \frac{\eta_k}{2} \|\nabla f(x_{k-1})\|^2 + \frac{\eta_k^2 L G^\alpha \tau^{2-\alpha}}{2} + \frac{\eta_k G^{2\alpha}}{2\tau^{2\alpha-2}} .$$

Rearrange and sum the terms above for some fixed step-size $\{\eta_k = \eta\}$ and threshold $\{\tau_k = \tau\}$ to get

$$\frac{1}{K} \sum_{k=1}^{K} \mathbb{E}\left[\|\nabla f(x_{k-1})\|^2\right] \leq \frac{2}{\eta K} (f(x_0) - \mathbb{E}[f(x_K)]) + \eta L G^\alpha \tau^{2-\alpha} + \frac{G^{2\alpha}}{\tau^{2\alpha-2}}$$

$$\leq \underbrace{\frac{2}{\eta K} (f(x_0) - f^\star)}_{T_1} + \underbrace{\eta L G^\alpha \tau^{2-\alpha} + \frac{G^{2\alpha}}{\tau^{2\alpha-2}}}_{T_2} .$$

Since we use a threshold $\tau = \frac{G}{(\eta L)^{\frac{1}{\alpha}}}$, we can simplify $T_2$ as

$$\eta L G^\alpha \tau^{2-\alpha} + \frac{G^{2\alpha}}{\tau^{2\alpha-2}} = \frac{2G^{2\alpha}}{\tau^{2\alpha-2}} = 2G^2 (\eta L)^{\frac{2\alpha-2}{\alpha}} .$$

Denote $f_0 = f(x_0) - f^\star$ to ease notation. Then, adding $T_2$ back to $T_1$ and using a step-size $\eta = \left[\left(\frac{f_0}{G^2 K}\right)^{\frac{\alpha}{3\alpha-2}} / L^{\frac{2\alpha-2}{3\alpha-2}}\right]$ we get

$$T_1 + T_2 = \frac{2f_0}{\eta K} + 2G^2 (\eta L)^{\frac{2\alpha-2}{\alpha}} = \frac{4f_0}{\eta K} = 4G^{\frac{2\alpha}{3\alpha-2}} \left(\frac{Lf_0}{K}\right)^{\frac{2\alpha-2}{3\alpha-2}} .$$

This proves the statement of the theorem. $\qquad\square$

## F    EFFECT OF COORDINATE-WISE MOMENT BOUND

We now examine how the rates would change if we replace Assumption 1 with Assumption 2.

### F.1    CONVERGENCE OF GCLIP (PROOF OF COROLLARY 1)

We now look at(GClip) under assumption 2. We can in fact also prove the following corollary for the **non-convex** case:

**Corollary 3** (**Non-convex GClip under coordinate-wise noise**). *Suppose we run GClip under the Assumption of 1 to obtain the sequence $\{x_k\}$. Then, if f is L-smooth and possibly non-convex, with appropriate step-sizes and averaging,*

$$\frac{1}{K} \sum_{k=1}^{K} \mathbb{E}[\|\nabla f(x_{k-1})\|^2] \leq 4(\sqrt{d}\|B\|_\alpha)^{\frac{2\alpha}{3\alpha-2}} \left(\frac{f(x_0) - f(x^\star)}{K}\right)^{\frac{2\alpha-2}{3\alpha-2}} .$$

The proof of both the convex and non-convex rates following directly from the following Lemma.

**Lemma 4.** *For any $g(x)$ suppose that assumption 2 with $\alpha \in (1, 2]$. Then suppose we have a constant upper-bound*

$$\mathbb{E}[\|g(x)\|^{\alpha}] \leq D.$$

*Then $D$ satisfies*

$$d^{\frac{\alpha}{2}-1}\|B\|^{\alpha}_{\alpha} \leq D \leq d^{\alpha/2}\|B\|^{\alpha}_{\alpha}.$$

*Proof.* Note that the function $(\cdot)^{\alpha/2}$ is concave for $\alpha \in (1, 2]$. Using Jensen's inequality we can rewrite as:

$$D \geq \mathbb{E}[\|g(x)\|^{\alpha}] = d^{\alpha/2}\mathbb{E}\left[\left(\frac{1}{d}\sum_{i=1}^{d}|g(x)^{(i)}|^2\right)^{\alpha/2}\right] \geq d^{\alpha/2-1}\mathbb{E}\left[\sum_{i=1}^{d}|g(x)^{(i)}|^{\alpha}\right].$$

Since the right hand-side can be as large as $d^{\frac{\alpha}{2}-1}\|B\|^{\alpha}_{\alpha}$, we have our first inequality. On the other hand, we also have an upper bound below:

$$\mathbb{E}[\|g(x)\|^{\alpha}] = \mathbb{E}\left[\left(\sum_{i=1}^{d}|g(x)^{(i)}|^2\right)^{\alpha/2}\right] \leq \mathbb{E}\left[\left(d(\max_{i=1}^{d}g(x)^{(i)})^2\right)^{\alpha/2}\right]$$

$$\leq \mathbb{E}\left[d^{\alpha/2}(\max_{i=1}^{d}g(x)^{(i)})^{\alpha}\right] \leq \mathbb{E}\left[d^{\alpha/2}\sum_{i=1}^{d}(g(x)^{(i)})^{\alpha}\right] \leq d^{\alpha/2}\sum_{i=1}^{d}B_i^{\alpha}$$

where $\|B\|^{\alpha}_{\alpha} = \sum_{i=1}^{d}B_i^{\alpha}$. Thus, we have shown that

$$d^{\frac{\alpha}{2}-1}\|B\|^{\alpha}_{\alpha} \leq \mathbb{E}[\|g(x)\|^{\alpha}] \leq d^{\alpha/2}\|B\|^{\alpha}_{\alpha}.$$

We know that Jensen's inequality is tight when all the co-ordinates have equal values. This means that if the noise across the coordinates is linearly correlated the lower bound is tighter, whereas the upper bound is tighter if the coordinates depend upon each other in a more complicated manner or are independent of each other. $\square$

Substituting this bound on $G$ in Theorems 1 and 2 gives us our corollaries.

### F.2 Convergence of CClip (Proof of Theorem 3)

The proof relies on the key lemma which captures the bias-variance trade off under the new noise-assumption and coordinate-wise clipping.

**Lemma 5.** *For any $g(x)$ suppose that assumption 2 with $\alpha \in (1, 2]$ holds. Denote $g_i$ to be $i_{th}$ component of $g(x)$, $\nabla f(x)_i$ to be $i_{th}$ component of $\nabla f(x)$. Then the estimator $\hat{g}(x) = [\hat{g}_1; \cdots ; \hat{g}_d]$ from (CClip) with clipping parameter $\tau = [\tau_1; \tau_2; \cdots ; \tau_d]$ satisfies:*

$$\mathbb{E}\left[\|\hat{g}_i\|^2\right] \leq B_i^{\alpha}\tau_i^{2-\alpha} \text{ and } \|\mathbb{E}[\hat{g}_i] - \nabla f(x)_i\|^2 \leq B_i^{2\alpha}\tau_i^{-2(\alpha-1)}.$$

*Proof.* Apply Lemma 2 to the one dimensional case in each coordinate. $\square$

*Proof of Theorem 3.* Theorem 1 For simplicity, we denote $\hat{g}_k = \eta_k\hat{g}(x_k)$ and the bias $b_k = \mathbb{E}[\hat{g}_k] - \nabla f(x_k)$.

$$\begin{aligned}
\|x_k - x^*\|^2 &= \|x_{k-1} - \eta_k\hat{g}_{k-1} - x^*\|^2 \\
&= \|x_{k-1} - x^*\|^2 - 2\eta_k\langle x_{k-1} - x^*, \nabla f(x_{k-1})\rangle \\
&\quad - 2\eta_k\langle x_{k-1} - x^*, b_{k-1}\rangle + \eta_k^2\|\hat{g}_{k-1}\|^2 \\
&\leq (1 - \mu\eta_k)\|x_{k-1} - x^*\|^2 - 2\eta_k(f(x_{k-1}) - f^*)) \\
&\quad + 2\eta_k(\frac{\mu}{4}\|x_{k-1} - x^*\|^2 + \frac{4}{\mu}\|b_k\|^2) + \eta_k^2\|\hat{g}_{k-1}\|^2.
\end{aligned}$$

Rearrange and we get

$$f(x_{k-1}) - f^* \leq \frac{\eta_k^{-1} - \mu/2}{2}\|x_{k-1} - x^*\|^2 - \frac{\eta_k^{-1}}{2}\|x_k - x^*\|^2 + \frac{4}{\mu}\|b_k\|^2 + \frac{\eta_k}{2}\|\hat{g}_{k-1}\|^2.$$

After taking expectation and apply the inequality from Lemma 2, we get

$$\mathbb{E}[f(x_{k-1})] - f^* \leq \mathbb{E}\left[\frac{\eta_k^{-1} - \mu/2}{2}\|x_{k-1} - x^*\|^2 - \frac{\eta_k^{-1}}{2}\|x_k - x^*\|^2\right]$$
$$+ \sum_{i=1}^d 4B_i^{2\alpha}\tau_i^{2-2\alpha}\mu^{-1} + \eta_k G^\alpha \tau_i^{2-\alpha}/2.$$

Then take $\eta_k = \frac{4}{\mu(k+1)}, \tau_i = B_i k^{\frac{1}{\alpha}}$ and multiply both side by $k$, we get

$$k\mathbb{E}[f(x_{k-1})] - f^* \leq \frac{\mu}{8}\mathbb{E}\left[k(k-1)\|x_{k-1} - x^*\|^2 - k(k+1)\|x_k - x^*\|^2\right]$$
$$+ 8\sum_{i=1}^d B_i^2 k^{\frac{2-\alpha}{\alpha}}\mu^{-1}.$$

Notice that $\sum_{k=1}^K k^{\frac{2-\alpha}{\alpha}} \leq \int_0^{K+1} k^{\frac{2-\alpha}{\alpha}} dk \leq (K+1)^{2/\alpha}$. Sum over $k$ and we get

$$\sum_{k=1}^K k\mathbb{E}[f(x_{k-1})] - f^* \leq \frac{\mu}{8}\mathbb{E}\left[-T(T+1)\|x_T - x^*\|^2\right]$$
$$+ 8\sum_{i=1}^d B_i^2 k^{\frac{2-\alpha}{\alpha}}\mu^{-1}.$$

Devide both side by $\frac{K(K+1)}{2}$ and we get

$$\frac{2}{K(K+1)}\sum_{k=1}^K k\mathbb{E}[f(x_{k-1})] - f^* \leq 8\sum_{i=1}^d B_i^2 k^{\frac{2-\alpha}{\alpha}}\mu^{-1}.$$

Notice that for $K \geq 1$, $K^{-1} \leq 2(K+1)^{-1}$. We have

$$\frac{2}{K(K+1)}\sum_{k=1}^K k\mathbb{E}[f(x_{k-1})] - f^* \leq 16\sum_{i=1}^d B_i^2 k^{\frac{2-\alpha}{\alpha}}\mu^{-1}.$$

The theorem then follows by Jensen's inequality. □ □

We can in fact also show a **non-convex** convergence rate:

**Theorem 5 (Non-convex CClip under coordinate-wise noise).** *Suppose we run CClip under the Assumption of 1 to obtain the sequence $\{x_k\}$. Then, if $f$ is L-smooth and possibly non-convex, with appropriate step-sizes and averaging,*

$$\frac{1}{K}\sum_{k=1}^K \mathbb{E}[\|\nabla f(x_{k-1})\|^2] \leq 4\|B\|_2^{\frac{2\alpha}{3\alpha-2}}\left(\frac{f(x_0) - f(x^\star)}{K}\right)^{\frac{2\alpha-2}{3\alpha-2}}.$$

*Proof.* This is very similar to Theorem 2, except that we use the same trick as in the proof above for Theorem 3. □

## G    LOWER BOUND (PROOF OF THEOREM 4)

We consider the following simple one-dimensional function class parameterized by $b$:

$$\min_{x \in [0,1/2]}\left\{f_b(x) = \tfrac{1}{2}(x-b)^2\right\}, \text{ for } b \in [0, 1/2]. \quad (1)$$

Also suppose that for $\alpha \in (1, 2]$ and $b \in [0, 1/2]$ the stochastic gradients are of the form:

$$g(x) \sim \nabla f_b(x) + \chi_b, \mathbb{E}[g(x)] = \nabla f_b(x), \text{ and } \mathbb{E}[|g(x)|^\alpha] \leq 1. \quad (2)$$

Note that the function class (1) has $\mu = 1$ and optimum value $f_b(b) = 0$, and the $\alpha$-moment of the noise in (2) satisfies $G = B \leq 1$. Thus, we want to prove the following:

**Theorem 6.** *For any $\alpha \in (1, 2]$ there exists a distribution $\chi_b$ such that the stochastic gradients satisfy* (2). *Further, for any (possibly randomized) algorithm $\mathcal{A}$, define $\mathcal{A}_k(f_b + \chi_b)$ to be the output of the algorithm $\mathcal{A}$ after $k$ queries to the stochastic gradient $g(x)$. Then:*

$$\max_{b \in [0, 1/2]} \mathbb{E}[f_b(\mathcal{A}_k(f_b + \chi_b))] \geq \Omega\left(\frac{1}{k^{2(\alpha-1)/\alpha}}\right).$$

Our lower bound construction is inspired by Theorem 2 of Bubeck et al. (2013). Let $\mathcal{A}_k(f_b + \chi_b)$ denote the output of any possibly randomized algorithm $\mathcal{A}$ after processing $k$ stochastic gradients of the function $f_b$ (with noise drawn i.i.d. from distribution $\chi_b$). Similarly, let $\mathcal{D}_k(f_b + \chi_b)$ denote the output of a *deterministic* algorithm after processing the $k$ stochastic gradients. Then from Yao's minimax principle we know that for any fixed distribution $\mathcal{B}$ over $[0, 1/2]$,

$$\min_{\mathcal{A}} \max_{b \in [0, 1/2]} \mathbb{E}_{\mathcal{A}}[\mathbb{E}_{\chi_b} f_b(\mathcal{A}_k(f_b + \chi_b))] \geq \min_{\mathcal{D}} \mathbb{E}_{b \sim \mathcal{B}}[\mathbb{E}_{\chi_b} f_b(\mathcal{D}_k(f_b + \chi_b))].$$

Here we denote $\mathbb{E}_{\mathcal{A}}$ to be expectation over the randomness of the algorithm $\mathcal{A}$ and $\mathbb{E}_{\chi_b}$ to be over the stochasticity of the the noise distribution $\chi_b$. Hence, we only have to analyze deterministic algorithms to establish the lower-bound. Further, since $\mathcal{D}_k$ is deterministic, for any *bijective* transformation $h$ which transforms the stochastic gradients, there exists a deterministic algorithm $\tilde{\mathcal{D}}$ such that $\tilde{\mathcal{D}}_k(h(f_b + \chi_b)) = \mathcal{D}_k(f_b + \chi_b)$. This implies that for any bijective transformation $h(\cdot)$ of the gradients:

$$\min_{\mathcal{D}} \mathbb{E}_{b \sim \mathcal{B}}[\mathbb{E}_{\chi_b} f_b(\mathcal{D}_k(f_b + \chi_b))] = \min_{\mathcal{D}} \mathbb{E}_{b \sim \mathcal{B}}[\mathbb{E}_{\chi_b} f_b(\mathcal{D}_k(h(f_b + \chi_b)))].$$

In this rest of the proof, we will try obtain a lower bound for the right hand side above.

We now describe our construction of the three quantities to be defined: the problem distribution $\mathcal{B}$, the noise distribution $\chi_b$, and the bijective mapping $h(\cdot)$. All of our definitions are parameterized by $\alpha \in (1, 2]$ (which is given as input) and by $\epsilon \in (0, 1/8)$ (which represents the desired target accuracy). We will pick $\epsilon$ to be a fixed constant which depends on the problem parameters (e.g. $k$) and should be thought of as being small.

- Problem distribution: $\mathcal{B}$ picks $b_0 = 2\epsilon$ or $b_1 = \epsilon$ at random i.e. $\nu \in \{0, 1\}$ is chosen by an unbiased coin toss and then we pick

$$b_\nu = (2 - \nu)\epsilon. \tag{3}$$

- Noise distribution: Define a constant $\gamma = (4\epsilon)^{1/(\alpha-1)}$ and $p_\nu = (\gamma^\alpha - 2\nu\gamma\epsilon)$. Simple computations verify that $\gamma \in (0, 1/2)$ and that

$$p_\nu = (4\epsilon)^{\frac{\alpha}{\alpha-1}} - 2\nu(4\epsilon^\alpha)^{\frac{1}{\alpha-1}} = (4 - 2\nu)(4\epsilon^\alpha)^{\frac{1}{\alpha-1}} \in (0, 1).$$

Then, for a given $\nu \in \{0, 1\}$ the stochastic gradient $g(x)$ is defined as

$$g(x) = \begin{cases} x - \frac{1}{2\gamma} & \text{with prob. } p_\nu, \\ x & \text{with prob. } 1 - p_\nu. \end{cases} \tag{4}$$

To see that we have the correct gradient in expectation verify that

$$\mathbb{E}[g(x)] = x - \frac{p_\nu}{2\gamma} = x - \frac{\gamma^{\alpha-1}}{2} + \nu\epsilon = x - (2 - \nu)\epsilon = x - b_\nu = \nabla f_{b_\nu}(x).$$

Next to bound the $\alpha$ moment of $g(x)$ we see that

$$\mathbb{E}[|g(x)|^\alpha] \leq \gamma^\alpha \left(x - \frac{1}{2\gamma}\right)^\alpha + x^\alpha \leq \frac{1}{2} + \frac{1}{2} = 1.$$

The above inequality used the bounds that $\alpha \geq 1$, $x \in [0, 1/2]$, and $\gamma \in (0, 1/2)$. Thus $g(x)$ defined in (4) satisfies condition (2).

- Bijective mapping: Note that here the only unknown variable is $\nu$ which only affects $p_\nu$. Thus the mapping is bijective as long as the *frequencies* of the events are preserved. Hence given a stochastic gradient $g(x_i)$ the mapping we use is:

$$h(g(x_i)) = \begin{cases} 1 & \text{if } g(x_i) = x_i - \frac{1}{2\gamma}, \\ 0 & \text{otherwise.} \end{cases} \tag{5}$$

Given the definitions above, the output of algorithm $\mathcal{D}_k$ is thus simply a function of $k$ i.i.d. samples drawn from the Bernoulli distribution with parameter $p_\nu$ (which is denoted by $\text{Bern}(p_\nu)$). We now show how achieving a small optimization error implies being able to guess the value of $\nu$.

**Lemma 6.** *Suppose we are given problem and noise distributions defined as in* (3) *and* (4)*, and an bijective mapping $h(\cdot)$ as in* (5)*. Further suppose that there is a deterministic algorithm $\mathcal{D}_k$ whose output after processing $k$ stochastic gradients satisfies*

$$\mathbb{E}_{b \sim \mathcal{B}}[\mathbb{E}_{\chi_b} f_b(\mathcal{D}_k(h(f_b + \chi_b)))] < \epsilon^2/64 \,.$$

*Then, there exists a deterministic function $\tilde{\mathcal{D}}_k$ which given $k$ independent samples of $\text{Bern}(p_\nu)$ outputs $\nu' = \tilde{\mathcal{D}}_k(\text{Bern}(p_\nu)) \in \{0, 1\}$ such that*

$$\Pr\left[\tilde{\mathcal{D}}_k(\text{Bern}(p_\nu)) = \nu\right] \geq \frac{3}{4} \,.$$

*Proof.* Suppose that we are given access to $k$ samples of $\text{Bern}(p_\nu)$. Use these $k$ samples as the input $h(f_b + \chi_b))$ to the procedure $\mathcal{D}_k$ (this is valid as previously discussed), and let the output of $\mathcal{D}_k$ be $x_k^{(\nu)}$. The assumption in the lemma states that

$$\mathbb{E}_\nu\left[\mathbb{E}_{\chi_b}|x_k^{(\nu)} - b_\nu|^2\right] < \frac{\epsilon^2}{32}, \text{ which implies that } \mathbb{E}_{\chi_b}|x_k^{(\nu)} - b_\nu|^2 < \frac{\epsilon^2}{16} \text{ almost surely.}$$

Then, using Markov's inequality (and then taking square-roots on both sides) gives

$$\Pr\left[|x_k^{(\nu)} - b_\nu| \geq \frac{\epsilon}{2}\right] \leq \frac{1}{4} \,.$$

Consider a simple procedure $\tilde{\mathcal{D}}_k$ which outputs $\nu' = 0$ if $x_k^{(\nu)} \geq \frac{3\epsilon}{2}$, and $\nu' = 1$ otherwise. Recall that $|b_0 - b_1| = \epsilon$ with $b_0 = 2\epsilon$ and $b_1 = \epsilon$. With probability $\frac{3}{4}$, $|x_k^{(\nu)} - b_\nu| < \frac{\epsilon}{2}$ and hence the output $\nu'$ is correct. $\qquad\square$

Lemma 6 shows that if the optimization error of $\mathcal{D}_k$ is small, there exists a procedure $\tilde{\mathcal{D}}_k$ which distinguishes between the Bernoulli distributions with parameters $p_0$ and $p_1$ using $k$ samples. To argue that the optimization error is large, one simply has to argue that a large number of samples are required to distinguish between $\text{Bern}(p_0)$ and $\text{Bern}(p_1)$.

**Lemma 7.** *For any deterministic procedure $\tilde{\mathcal{D}}_k(\text{Bern}(p_\nu))$ which processes $k$ samples of $\text{Bern}(p_\nu)$ and outputs $\nu'$*

$$\Pr[\nu' = \nu] \leq \frac{1}{2} + \sqrt{k(4\epsilon)^{\frac{\alpha}{\alpha-1}}} \,.$$

*Proof.* Here it would be convenient to make the dependence on the samples explicit. Denote $s_k^{(\nu)} = \left(s_1^{(\nu)}, \dots, s_k^{(\nu)}\right) \in \{0, 1\}^k$ to be the $k$ samples drawn from $\text{Bern}(p_\nu)$ and denote the output as $\nu' = \tilde{\mathcal{D}}(s_k^{(\nu)})$. With some slight abuse of notation where we use the same symbols to denote the realization and their distributions, we have:

$$\Pr\left[\tilde{\mathcal{D}}(s_k^{(\nu)}) = \nu\right] = \frac{1}{2}\Pr\left[\tilde{\mathcal{D}}(s_k^{(1)}) = 1\right] + \frac{1}{2}\Pr\left[\tilde{\mathcal{D}}(s_k^{(0)}) = 0\right] = \frac{1}{2} + \frac{1}{2}\mathbb{E}\left[\tilde{\mathcal{D}}(s_k^{(1)}) - \tilde{\mathcal{D}}(s_k^{(0)})\right] \,.$$

Next using Pinsker's inequality we can upper bound the right hand side as:

$$\mathbb{E}\left[\tilde{\mathcal{D}}(s_k^{(1)}) - \tilde{\mathcal{D}}(s_k^{(0)})\right] \leq \left|\tilde{\mathcal{D}}(s_k^{(1)}) - \tilde{\mathcal{D}}(s_k^{(0)})\right|_{TV} \leq \sqrt{\frac{1}{2}\text{KL}\left(\tilde{\mathcal{D}}\left(s_k^{(1)}\right), \tilde{\mathcal{D}}\left(s_k^{(0)}\right)\right)} \,,$$

where $|\cdot|_{TV}$ denotes the total-variation distance and $\text{KL}(\cdot, \cdot)$ denotes the KL-divergence. Recall two properties of KL-divergence: i) for a product measures defined over the same measurable space $(p_1, \dots, p_k)$ and $(q_1, \dots, q_k)$,

$$\text{KL}((p_1, \dots, p_k), (q_1, \dots, q_k)) = \sum_{i=1}^{k} \text{KL}(p_i, q_i) \,,$$

and ii) for any deterministic function $\tilde{\mathcal{D}}$,

$$\mathrm{KL}(p, q) \geq \mathrm{KL}(\tilde{\mathcal{D}}(p), \tilde{\mathcal{D}}(q)) \,.$$

Thus, we can simplify as

$$\Pr\left[\tilde{\mathcal{D}}(s_k^{(\nu)}) = \nu\right] \leq \frac{1}{2} + \sqrt{\frac{k}{8}\,\mathrm{KL}(\mathrm{Bern}(p_1), \mathrm{Bern}(p_0))}$$

$$\leq \frac{1}{2} + \sqrt{\frac{k}{8}\frac{(p_0 - p_1)^2}{p_0(1 - p_0)}}$$

$$\leq \frac{1}{2} + \sqrt{\frac{k(\gamma\epsilon)^2}{4\gamma^\alpha}}$$

$$= \frac{1}{2} + \sqrt{k\left(4^{(2-1/\alpha)}\epsilon\right)^{\frac{\alpha}{\alpha-1}}} \,.$$

Recalling that $\alpha \in (1, 2]$ gives us the statement of the lemma. $\qquad\square$

If we pick $\epsilon$ to be

$$\epsilon = \frac{1}{16k^{(\alpha-1)/\alpha}} \,,$$

we have that

$$\frac{1}{2} + \sqrt{k(4\epsilon)^{\frac{\alpha}{\alpha-1}}} < \frac{3}{4} \,.$$

Given Lemmas 6 and 7, this implies that for the above choice of $\epsilon$,

$$\mathbb{E}_{b\sim\mathcal{B}}[\mathbb{E}_{\chi_b} f_b(\mathcal{D}_k(h(f_b + \chi_b)))] \geq \epsilon^2/64 = \frac{1}{2^{14}k^{2(\alpha-1)/\alpha}} \,.$$

This finishes the proof of the theorem. Note that the readability of the proof was prioritized over optimality and it is possible to obtain significantly better constants. $\qquad\square$

## H   A COMPARISON WITH (SIMSEKLI ET AL., 2019)

We are not the first to study the heavy-tailed noise behavior in neural network training. The novel work by Simsekli et al. (2019) studies the noise behavior of AlexNet on Cifar 10 and observed that the noise does not seem to come from Gaussian distribution. However, in our AlexNet training with ImageNet data, we observe that the noise histogram looks Gaussian as in Figure 6(a, b). We believe the difference results from that in (Simsekli et al., 2019), the authors treat the noise in each coordinate as an independent scaler noise, as described in the original work on applying tail index estimator. We on the other hand, consider each the noise as a high dimensional random vector computed from a minibatch. We are also able to observe heavy tailed noise if we fix a single minibatch and plot the noise in each dimension, as shown in Figure 6(c). The fact that noise is well concentrated on Cifar is also observed by Panigrahi et al. (2019).

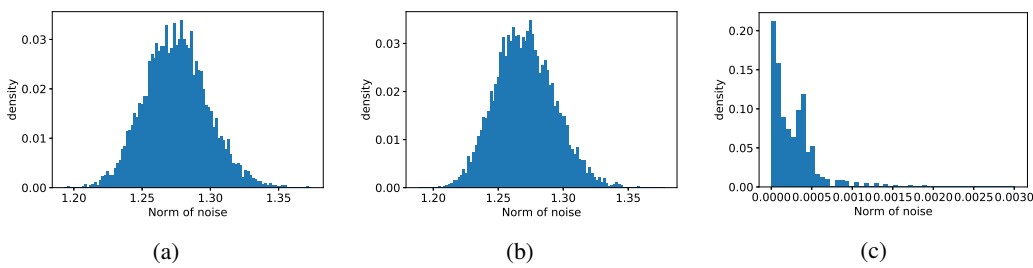

Figure 6: (a) Noise histogram of AlexNet on ImageNet data at initialization. (b)Noise histogram of AlexNet on ImageNet data at 5k iterations. (c) The per dimension noise distribution within a single minibatch at initialization.

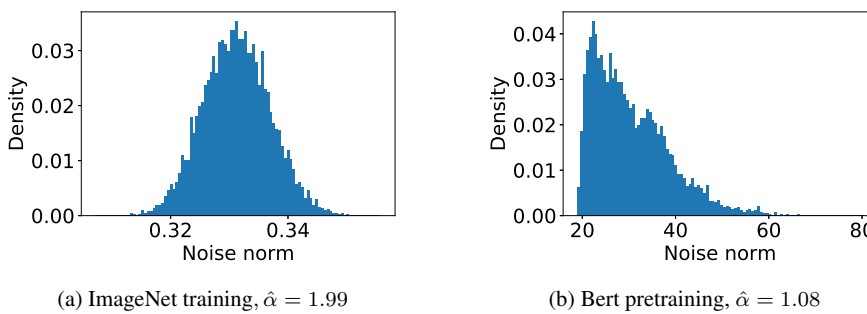

(a) ImageNet training, $\hat{\alpha} = 1.99$    (b) Bert pretraining, $\hat{\alpha} = 1.08$

Figure 7: Tail index estimation of gradient noise in ImageNet training and BERT training.

Furthermore, we used the tail index estimator presented in Simsekli et al. (2019) to estimate the tail index of noise norm distribution. Though some assumptions of the estimator are not satisfied (in our case, the symmetry assumption; in Simsekli et al. (2019), the symmetry assumption and independence assumption), we think it can be an indicator for measuring the "heaviness" of the tail distribution.

# I   ACCLIP IN IMAGENET TRAINING

For completeness, we test ACClip on ImageNet training with ResNet50. After hyperparameter tuning for all algorithms, ACClip is able to achieve better performance compared to ADAM, but worse performance compared to SGD. This is as expected because the noise distribution in ImageNet + ResNet50 training is well concentrated. The validation accuracy for SGD, ADAM, ACClip are $0.754, 0.716, 0.730$ respectively.

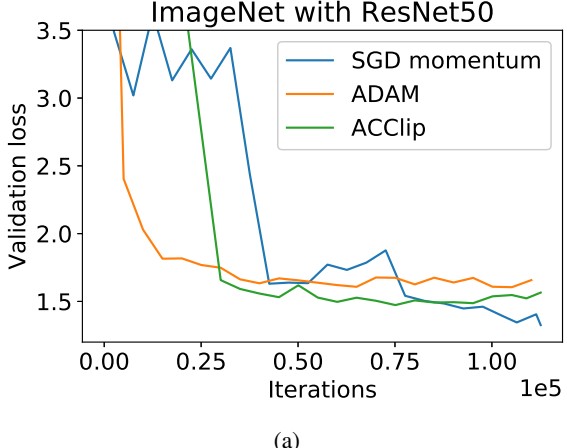

(a)

Figure 8: Validation loss for ResNet50 trained on ImageNet. SGD outperforms Adam and ACClip.

