# OpenReview forum: "Why ADAM Beats SGD for Attention Models	"
_ICLR.cc/2020/Conference — Reject_

### Official Review · AnonReviewer1 · 2019-10-21
**Official Blind Review #1**

**Rating:** 3

**Review:**

This paper gives theoretical and empirical results for a gradient clipping variant of Adam they call ACClip.  While the theoretical analysis is rather  sophisticated and nontrivial, I personally do not believe that analyses of this form are of any value in guiding practice.  But that is a long discussion that is not specific to this paper.  The bottom line is that for me it is mainly the experimental results that matter.

The experimental results are not compelling.  It is now clear that careful hyperparameter search is critical to drawing experimental conclusions about optimizers.  This paper simply states the hyperparameters used with no discussion of hyperparameter search. I strongly believe that any claim about optimizers needs to be backed up by experiments with very careful hyper-parameter optimization.

Postscript:  I have modified this review in response to the authors.  I remain unconvinced that the theory is providing anything more than an intuitive hypothesis that Adam is importance when the variance is large.  Since Adam and RMSprop are explicitly damping variance in the gradients, this intuition is reasonable even before we prove any theorems.  I still believe the theorems do not add really add anything to the intuition and it is the experiments that matter.

**Experience Assessment:**

I have read many papers in this area.

**Review Assessment: Checking Correctness Of Derivations And Theory:**

I assessed the sensibility of the derivations and theory.

**Review Assessment: Checking Correctness Of Experiments:**

I assessed the sensibility of the experiments.

**Review Assessment: Thoroughness In Paper Reading:**

I read the paper at least twice and used my best judgement in assessing the paper.

---

> ### Author Response · Authors · 2019-11-08
> **We propose a theoretical analysis that aligns with empirical observations**
>
> We thank the reviewer for the comments and feedback. We address the reviewer’s questions as follows:
>
> 1“analyses of this form are of no value in guiding practice;  it is mainly the experimental results that matter”:
>
> The discrepancy between convergence analysis and empirical result is the main motivation of our work. Particularly, we show that under a more realistic assumption that noise is heavy tailed, theory can guide practice.
>
> We would like to emphasize that the central goal of the paper is to address the important practical question: Why Adam outperforms SGD in Attention models? It is indeed puzzling why training these models necessitates the use of adaptive optimization techniques like Adam in comparison to computer vision models like ResNet where SGD + Momentum gives the best performance.
>
> In this paper, we hypothesize that heavy-tail noise is one root cause of this difference and provide strong theoretical (see Table 1 and corresponding theorems) and empirical evidence for this hypothesis. These results show that under heavy tail noise of stochastic gradients (or high variance in general) , the performance of SGD deteriorates significantly (see Assumption 1 and Section A of Appendix). While we believe that the convergence rates are interesting and important, even from a practical standpoint, our theoretical results provide qualitative insights into why SGD fails to perform well in the aforementioned settings.  Our analysis provides guidelines for the development of optimization techniques for attention models (indeed, ACClip follows as a direct consequence of this analysis).
>
>
>
> 2. “There is no mention of RoBERTa and her descendents.”:
>
> Since RoBERTa is a different training procedure (more data, more iterations, larger batch) of the same BERT model, we believe that ACClip should perform similarly well in these models. We will be happy to run more experiments and include them in the final version if this is the point of concern.
>
> We indeed extensively tuned the hyperparameters to get the best result for each optimizer and will include the details in the next version. We thank the reviewer for pointing out our oversight. In particular, we found that the set of ADAM hyperparameters used in the original BERT [Devlin et al] paper works the best. Hence we use the same params as in [Devlin et al] and achieved slightly better baseline performance.
>
> It will indeed be great if GLUE leader board members can examine the paper and try our optimization technique. We will be happy to help them in this process. That said, we would like to emphasize that the primary focus of the paper is not to achieve SOTA results for attention models but rather understand the optimization challenges in training attention models and provide guidelines for development of optimization techniques specially catered to these settings.

---

### Official Review · AnonReviewer3 · 2019-10-23
**Official Blind Review #3**

**Rating:** 6

**Review:**

The paper proposed a very interesting claim: When training a neural network, if the (stochastic) gradient noise is Gaussian-like, then SGD performs better than Adam; On the other hand if the gradient noise is Heavy tailed, then Adam perform better than SGD.

The paper supported this argument with experiments showing that ResNet50 on ImageNet, the noise is more like Gaussian while BERT on language learning tasks the noise is more heavy-tailed. The paper also gave a theoretical result showing that Adam converges in the regime of heavy-tailed noise.


The experiment finding is quite surprising to me, since many papers (see e.g.
A Tail-Index Analysis of Stochastic Gradient Noise in Deep Neural Networks
)  claim that the SGD noise is heavy-tailed for image recognization tasks such as CIFAR-10, CIFAR-100. The referred paper used rigorous statistical testing for the tail-index of the SGD noise, while this paper simply drew some image.


Moreover, the theoretical result in this paper also worries me quite a bit, since from the bounds it seems that Adam is the dominating algorithm (both in the heavy-tail case and in Gaussian tail case). Moreover, SGD also converges in the heavy-tail noise case (by showing that the norm of x_t is not too large during the training process using martingale-based argument). Hence, the upper bound of the theoretical result is convincing enough to claim that Adam is better than SGD in certain regime.


After Rebuttal: I have read the authors' responses and acknowledge the sensibility of the statement. I have higher my score: In particular, if the noise is indeed Gaussian as opposite to the "known results", this paper should be accepted.




**Experience Assessment:**

I have published in this field for several years.

**Review Assessment: Checking Correctness Of Derivations And Theory:**

I assessed the sensibility of the derivations and theory.

**Review Assessment: Checking Correctness Of Experiments:**

I assessed the sensibility of the experiments.

**Review Assessment: Thoroughness In Paper Reading:**

I read the paper at least twice and used my best judgement in assessing the paper.

---

> ### Author Response · Authors · 2019-11-08
> **Statistical test added and explained; ADAM is not dominating; SGD without clipping does not converge**
>
> We thank the reviewer for the comments and feedback. We address the reviewer’s questions as follows:
>
> 1. Gaussianity of Resnet noise: (Simsekli et al. 2019) plot the individual coordinates of the noise vector, whereas we examine the norm of the noise vector. The latter is more relevant for the convergence of SGD. Replicating the tail-index tests of (Simsekli et al. 2019) on the norm of the noise vectors confirms that Resnet is indeed gaussian (has alpha ≈ 2) and that BERT is heavy-tailed (has alpha ≈ 1). We refer to Appendix H of the updated version for details. Other recent papers (e.g. Panigrahi et al. 2019 “Non-Gaussianity of stochastic gradient noise”) also dispute the validity of the results in (Simsekli et al. 2019) due to estimating tail index of per-coordinate noise as a scalar random variable instead of the norm of a vector valued variable. Their results corroborate with our findings that Resnet has Gaussian noise.
>
>
> 2. Convergence of SGD with heavy tailed noise: Our theory suggests that whenever there is heavy-tailed noise, Adam will outperform SGD. In such situations, the variance is potentially infinite making the iterates of SGD unstable. The martingale concentration suggested by the reviewer requires sub-gaussian noise and cannot occur with heavy-tailed noise (see Appendix A for a formal treatment). Note that Adam (and Acclip) do not have this issue and converge even with heavy-tailed noise.
>
> 3. Convergence with sub-gaussian noise: On the other hand, if the noise is sub-gaussian then indeed SGD converges and the martingale concentration mentioned by the reviewer kicks in. In such cases, Adam and Acclip do not have any advantage over SGD. Thus, our theory perfectly reflects our experimental results and we see no discrepancy.

---

### Official Review · AnonReviewer2 · 2019-10-24
**Official Blind Review #2**

**Rating:** 6

**Review:**

This paper demonstrates empirically that the gradient noises of SGD with ResNet and Adam with Bert are different: one is well-concentrated, while the other one is heavy-tailed. The paper claims that this difference costs the failure of SGD on training Bert. Furthermore, the authors proposes gradient clipped SGD and its adaptive version ACClip. Experiments show that ACClip outperforms Adam on training Bert.

In general, the paper is well-written and has addressed an important practical and theoretical problem of why SGD fails to train Bert and how to fix this problem. The theory appears to be solid. My only concern is how generalizable ACClip is. Experiments show that it outperforms Adam on training Bert. How about the other architectures where Adam is usually applied? Is ACClip competitive to Adam in those applications? What’s the performance of ACClip on DL applications where SGD + momentum works well, such as ResNet on the ImageNet dataset?

What is exactly \delta f(x)? Is this the full batch gradient over all training examples?

Some typos:
1.	Page 1: thereby providing a explanation
2.	Page 4: at most af factor of 2 and Adam


**Experience Assessment:**

I do not know much about this area.

**Review Assessment: Checking Correctness Of Derivations And Theory:**

I assessed the sensibility of the derivations and theory.

**Review Assessment: Checking Correctness Of Experiments:**

I assessed the sensibility of the experiments.

**Review Assessment: Thoroughness In Paper Reading:**

I read the paper at least twice and used my best judgement in assessing the paper.

---

> ### Author Response · Authors · 2019-11-08
> **ImageNet experiments updated in the draft; LSTM experiments will be added.**
>
> We would like to thank the reviewer for the feedback and comments. We address the reviewer’s questions as follows:
>
> 1. Generalizability of ACClip:
>
> The development of ACClip followed from our analysis that heavy-tail noise is one root cause of the difference between SGD and Adam performance for Attention models. As such, we expect the performance of ACClip to generalize to other scenarios with heavy tail noise of stochastic gradients or high variance in general.
> We are happy to extend the experimental results so as to evaluate and highlight the generalizability of ACClip.
>
>          Regarding the Reviewer’s specific requests, we evaluated ACClip for ResNet on ImageNet and updated the draft (see Appendix I). However, we would like to emphasize that when gradient noise is concentrated like in Resnet, we do not claim that ACClip will perform better than other optimizers. Our focus is for models with heavy tailed noise like Transformers.
>          As another model where Adam outperforms SGD, we will also evaluate ACClip on an LSTM model for language modeling. While we are not sure if this will be ready by the end of the feedback period, we will definitely include this in the camera ready version of the paper.
>
> 2. Definition of  \delta f(x)
>
> It is the mean of the population gradient, i.e. expectation of the stochastic gradient when X is drawn from the population. This could be the full batch gradient if the objective is the empirical loss.

---

### Public Comment · ~Guodong_Zhang1 · 2019-10-03
**Heavy-tailed Gradient Noise**

Hi,

It's quite surprising to me that the gradient noise for ResNet is close to Gaussian.

As you mentioned in related works, Simsekli et al. (2019) is the first work to focus on this direction and they got different conclusions from you. If I remember right, they showed that the gradient noise is heavy-tailed for standard ConvNets on image classification tasks. The only difference I'm aware of is that they compute the noise of each parameter individually.

I wonder why the tail index of the whole gradient noise would be fundamentally different from that of the noise of a single parameter? Do you have any insight/explanation on that?

---

> ### Author Response · Authors · 2019-10-04
> **Thank you for the comment.**
>
>
> Hi Guodong,
>
> Thank you for your question.
>
> My understanding of this phenomenon is as follows:
>
> First, the definition of being "heavy-tailed" is a relative concept in real world problems. In other word, it's much easier to conclude that one distribution is more heavy-tailed than another, as opposed to define what it means for a distribution to be "heavy-tailed". In fact, with much more samples (by bootstrapping from the empirical distribution), we will finally observe the Gaussian behavior in all models with a finite dataset. However, the concentration will kick in much earlier for some model than for other models.
>
> Second, following the above argument, we only need to consider why the multidimensional gradient noise is better concentrated. This happens whenever the noise in each coordinate are not perfectly correlated. The gradient norm, which is squared root of sum of random variables will have some concentration due the independence or even negative correlation across coordinates. Due to the large number of parameters in a neural net, this concentration can lead to hugely different behavior.

---

### Author Response · Authors · 2019-11-08
**Draft updated**

We thank the reviewers for their comments that our work proposes a novel explanation for an important problem: why ADAM outperforms SGD in BERT pretraining. We updated our draft per reviewers’ requests with two changes.

First, we used tail-index estimator from [Simsekli et al] and confirmed our observation that BERT training has heavy-tailed noise but the noise in ImageNet training is well concentrated. A detailed comment is added to Appendix H. We would like to highlight that we are estimating the distribution of the norm of a vector-valued noise while [Simsekli et al] treated the noise in each coordinate as an independent scalar random variable. This leads to different conclusions.

Second, we tested the performance of ACClip on ImageNet training (added inAppendix I). The performance is worse than SGD but slightly better than Adam. This is consistent with our theory that with well-concentrated noise distribution, Adam-like algorithms do not outperform SGD.

We emphasize that our main contribution is a novel theoretically-justified explanation for why Adam-like algorithms outperforms SGD in certain tasks but not in others. It is not our goal to replace Adam. Our theoretical and experimental results serve to convincingly establish that the efficacy of Adam can be attributed to its adaptive clipping behavior and that it will outperform SGD whenever the noise is heavy-tailed.

---

### Decision · Program_Chairs · 2019-12-19

**Decision:**

Reject

**Comment:**

This paper tries to explain why Adam is better than sgd for training attention model. In specific, it first provides some empirical and theoretical evidence that a heavy-tailed distribution of the noise in stochastic gradients is the cause of SGD's worse performance. Then the authors studied a clipped variant of SGD that circumvents this issue, and revisited Adam through the lens of clipping. Overall, this paper conveys some interesting ideas. On the other hand, the theorems proved in this paper do not provide additional insight besides the intuition and the experiments are weak (hyperparameters are not carefully tuned). So even after author response, it still does not gather sufficient support from the reviewers. This is a borderline paper, and due to a rather limited number of papers the conference can accept, I encourage the authors to improve this paper and resubmit it to future conference.